# Identifying the Internal Coupling Coordination Relationship of Forest Ecological Security and Its Spatial Influencing Factors

Jiehua Lyu [1], Zhe Sun [1], Tingyu Yang [1], Bin Zhang [2] and Xiuting Cai [1,*]

1   College of Economics and Management, Northeast Forestry University, Harbin 150040, China; jiehualyu@nefu.edu.cn (J.L.); sunzhe0304@nefu.edu.cn (Z.S.); yangtingyu@nefu.edu.cn (T.Y.)
2   College of Business, Hebei Normal University, Shijiazhuang 050010, China; nikktt@nefu.edu.cn
*   Correspondence: caixiuting0120@nefu.edu.cn

**Abstract:** Forest ecological security is an important guarantee for national security and the healthy development of forestry. Existing research has been limited to the comprehensive evaluation and obstruction factors of forest ecological security, and this study innovatively analyzes internal coordination of forest ecological security and its spatial influencing factors in order to enhance the internal coordination of the forest ecosystem and promote the coordinated improvement of regional forest ecological security. Firstly, the forest ecological security in China from 2006 to 2020 was evaluated. On this basis, the coupling coordination degree of forest ecological security (FESD) was calculated, and its influencing factors and spatial spillover effects were analyzed using the spatial econometric model. The results show that: (1) most of China's provincial forest ecological security is at a critical and relatively safe level, and shows a trend of gradual improvement; (2) FESD in 25 provinces, represented by Guangxi, is in the acceptable range and is constantly improving; and (3) annual mean temperature, urbanization rate, completed forestry investment, and forest pest control have been positive influences on FESD. Forest population density, per capita GDP, and investment in environmental governance have significant negative influences. In addition, completed forestry investment, annual mean temperature, urbanization rate, forest population density, and forest pest control have significant spillover effects on FESD. Therefore, in the process of forest ecological management, it is necessary to further strengthen investments in forestry and pest control, and to pay more attention to the interaction between regions.

**Keywords:** forest ecological security; degree of coupling coordination; influencing factors; spatial spillover effect





## 1. Introduction

Forests are ecosystems with the largest areas on land. They can not only provide basic resource guarantees but also have the ecological functions of water conservation and biodiversity maintenance [1]. In the mid-1990s, due to the needs of national development and the planned economic regime, the production and supply of timber was the primary task of China's forestry [2,3]. Under this guidance, deforestation emerged as very widespread; the forest coverage rate decreased from 12.7 percent in the first National Forest Resources Inventory (1973–1976) to 12 percent in the second (1977–1981), and the structure of the forest species was seriously unbalanced, with timber forests accounting for more than 70% and shelterbelt forests accounting for less than 10% (the second National Forest Resources Inventory). As a result, ecological crises, such as species extinction, are frequent [4]. At the same time, with the increase in people's demand for forest resources and the expansion of woodland occupation, the stability of forest ecosystem is further destroyed [5]. In order to alleviate the plight of forest ecology, at the end of the last century, China successively implemented a series of forest protection policies, which promoted the balance of China's forest ecosystem [6,7]. However, China is still faced with the grim reality of low-level forest

ecological security and its obvious regional differences due to the long cycle of ecological restoration [8,9]. In fact, the problem of forest ecological security is the contradiction between economic development and forest ecological protection [10], that is, the coordination problem among the pressures of human production on the forest ecosystem, the state of the forest ecosystem, and the response strategies employed by humanity to protect the forest ecosystem. Hence, investigating the coordination between pressure, state, and response in forest ecological security from a system endogeny perspective offers a novel approach to enhancing forest ecological security, and it is the focus of this study.

Regarding the internal mechanism of forest ecological security, it encapsulates the interactive relationship between mankind and the forest ecosystem. Human beings obtain the forest resources from the forest ecosystem, and at the same time, they release pollutants into the ecological environment, consequently altering the quantity, quality, and well-being of forest resources. In turn, the change in the forest ecosystem restrains human socio-economic activities, and society responds to the change in the forest ecosystem through various policies.

The research on sustainable development of forest ecosystems at home and abroad involves many aspects, such as forest health, forest ecosystem services, and forest ecological security. Among them, forest health is a concept for evaluating forest resource management, which depends on the age, structure, disease, etc. of the forest [11], and is affected by many factors, such as the climate, humans, and biology [12]. As the deterioration of the forest ecological environment, a large number of forest health monitoring studies [13–17] and studies on the relationship between forest health and atmospheric environment have been carried out in many countries [18–21]. Ecosystem services refer to the well-being of ecosystems for humans [22]. Forests play an irreplaceable and significant role in provisioning diverse ecosystem services, and products that support both human well-being and the Earth's life support system [23,24]. People have gradually realized the importance of forest ecosystem service function for sustainable development. As a consequence, numerous studies have been conducted on the methodologies and applications involved in assessing the value of forest ecosystem services [25–28].

Research on the forest health center has focused on evaluating the condition of the forest ecosystem itself. The service function of forest ecosystems mainly refers to the tangible and intangible services which humans obtain from forest ecosystems. To a certain extent, forest ecological security is a combination of forest health and forest ecosystem services, and embodies the concept of harmonious coexistence between man and nature [29]. From the existing research, the study of forest ecological security mainly includes the establishment of the evaluation index system and a comprehensive assessment of forest ecological security. Specifically, the establishment of an index system is an important prerequisite for scientific evaluation of forest ecological security. Many experts construct the index system of forest ecological security based on PSR and DPSIR [30–32], and have used the fuzzy comprehensive evaluation method, the improved TOPSIS method, the comprehensive evaluation method, and the system dynamics model to comprehensively evaluate forest ecological security [29,33]. On this basis, these authors discuss the spatial distribution characteristics [8,34] and influencing factors [35,36]. In addition, some experts' research objects are the relationships between forest ecological security and forest product trade [37], forest management and protection [38], forestry industry structure [39], and forest food safety [40].

In conclusion, the majority of existing studies are concentrated on conducting comprehensive assessments of forest ecological security and endeavoring to enhance its condition by analyzing the factors influencing forest ecological security. It is rare to explore ways to improve forest ecological security from its internal coordination. Therefore, this study first builds a forest ecological security evaluation index system, and calculates the comprehensive index of forest ecological security. Building upon this foundation, the coupling coordination degree model is employed to analyze the developmental level of the pressure-state-response coupling coordination degree. Furthermore, the spatial econometric model

is used to analyze the influencing factors of the degree of coupling and coordination in order to provide suggestions for the improvement of forest ecological security from the perspective of the internal coordination of the system.

## 2. Materials and Methods

### 2.1. Constructing the Index System

The concept of forest ecological security can be categorized into two dimensions: a narrow and a broad sense. In the narrow sense, forest ecological security pertains solely to the state of the forest ecosystem, encompassing aspects such as integrity, health, and sustainability [41]. In the broad sense, forest ecological security denotes a condition wherein the ecological services furnished by the forest ecosystem can meet the sustainable requirements for humans, ensuring that human production and life will remain free from threats while maintaining the integrity of the structure and functioning of the forest ecosystem within a defined spatial-temporal scope [29,33].

This shows that the forest ecological security in the broad sense includes not only the resources and the health state of the forest ecosystem but also human intervention in the forest ecosystem. The types of intervention are divided into destructive interventions, such as timber harvesting, destroying forests, and occupied forest land, etc., and protective intervention, such as afforestation, reforestation, etc. Beginning from the generalized concept of forest ecological security, this study constructs an index system of forest ecological security from three aspects: pressure, state, and response, based on the pressure–state–response (PSR) logical framework [42] (Table 1).

Among these, pressure indicators are utilized to assess the adverse effects of human economic and social activities on the forest ecosystem, encompassing aspects such as the consumption of forest resources in production and daily life, the encroachment of forest land during urbanization, and the degradation of the forest growth environment due to pollutants arising from industrial production. State indicators are used to describe the quantity, quality, and health of forest resources in a specific period. Response indicators are used to characterize how society and individuals prevent and mitigate the adverse impact of human production and living activities on forest ecosystems as well as remedial measures for changes to the forest ecological environment that have occurred which are not conducive to human survival or development. The specific indicators are shown in Table 1.

### 2.2. Methods

#### 2.2.1. Comprehensive Evaluation

(1)   Data standardization

To mitigate the influence of dimensionality and attribute disparities on the evaluation outcomes, it is imperative to standardize the raw data of the indicators. In this study, the forest ecological security system is divided into three subsystems. $x_{ij}$ is the $j$th index in the $i$th subsystem, and $x_{ij\max}$ and $x_{ij\min}$ are the maximum and minimum values of the $j$th index in the $i$th subsystem, respectively. When $x_{ij}$ is larger, the system function is better, indicating that $x_{ij}$ is a positive indicator; otherwise, $x_{ij}$ is a negative indicator. Standardized formulae for different attribute indicators are as follows:

$$x_{ij}' = \frac{x_{ij} - x_{ij\min}}{x_{ij\max} - x_{ij\min}} \text{ (Positive indicator)} \tag{1}$$

$$x_{ij}' = \frac{x_{ij\max} - x_{ij}}{x_{ij\max} - x_{ij\min}} \text{ (Negative indicator)} \tag{2}$$

**Table 1.** Index system of forest ecological security.

| Level Indicators | The Secondary Indicators | Indicator Name (Unit) | Formula | Type of Indicator | Qualitative Weight | Quantitative Weight | Combination Weight |
|---|---|---|---|---|---|---|---|
| Forest ecological security (FES) | External pressures on forest ecosystems (P) | Intensity of human interference (%) | (Building area + Arable area)/Administrative area × 100% | − | 0.3826 | 0.3022 | 0.3424 |
| | | Total forestry output value (CNY ten thousand) | Obtained directly | − | 0.1588 | 0.2127 | 0.1858 |
| | | Intensity of land desertification (%) | Land desertification area/Administrative area × 100% | − | 0.1783 | 0.1463 | 0.1623 |
| | | Intensity of SO$_2$ (t/ha) | SO$_2$ volume/Administrative area | − | 0.0626 | 0.1176 | 0.0901 |
| | | Intensity of industrial wastewater (t/ha) | Industrial wastewater volume/Administrative area | − | 0.0946 | 0.0869 | 0.0908 |
| | | Intensity of solid waste (t/ha) | Solid waste volume/Administrative area | − | 0.1231 | 0.1343 | 0.1287 |
| | The state of forest ecosystems (S) | Forest coverage rate (%) | Forest area/Administrative area × 100% | + | 0.3418 | 0.3040 | 0.3229 |
| | | Forest stock volume per unit forest area (m$^3$/ha) | Forest stock volume/Forest area | + | 0.3056 | 0.3225 | 0.3140 |
| | | Proportion of natural forests (%) | Natural forest area/Forest area × 100% | + | 0.2016 | 0.2314 | 0.2165 |
| | | Proportion of forest fire(‰) | Area of forest fire/Forest area × 1000‰ | − | 0.0978 | 0.0707 | 0.0842 |
| | | Proportion of forest pests and diseases (%) | Area of forest diseases and pests/Forest area × 100% | − | 0.0532 | 0.0714 | 0.0623 |
| | Response measures of forest ecological protection (R) | Proportion of reforestation (%) | Reforestation area/Administrative area × 100% | + | 0.4467 | 0.3936 | 0.42013 |
| | | Proportion of mountain closure and forest cultivation (%) | Area to close mountains and cultivate forests/Administrative area × 100% | + | 0.4058 | 0.3874 | 0.3966 |
| | | Compliance rate of industrial wastewater (%) | The standard amount of industrial wastewater/Total industrial wastewater × 100% | + | 0.0693 | 0.1017 | 0.0855 |
| | | Utilization rate of industrial solid waste (%) | Industrial solid waste utilization/Total industrial solid waste × 100% | + | 0.0782 | 0.1173 | 0.0978 |

Notes: "+" represents a positive indicator, which has a positive role in promoting forest ecological security; "−" represents a negative indicator, which has an inhibitory effect on forest ecological security.

(2) Weight calculation

The determination of index weight has an important influence on the rationality of the results. In order to reflect the subjective and objective information comprehensively, we selected the analytic hierarchy process (AHP) [43] and the entropy weight method to determine the index weight comprehensively.

① The steps of AHP are as follows:

Firstly, a hierarchical analysis framework is established using the evaluation index system.

Secondly, the judgment matrix is constructed. This study introduces the 1–9 scale method, selects several experts in the field to assign values to each index, and constructs a comparison judgment matrix via pairwise comparison of the indicators in each level.

$$A = (a_{ij})_{m \times n} = \begin{pmatrix} a_{11} & \cdots & a_{1n} \\ \vdots & \ddots & \vdots \\ a_{m1} & \cdots & a_{mn} \end{pmatrix} \tag{3}$$

Thirdly, the index weight is calculated under the condition of a single criterion. The characteristic root method is applied to compute the weight coefficient of each index based on the judgment matrix, and normalized processing is carried out to obtain the weight of each index at a certain level relative to an index at the upper level. The calculation formula is as follows:

$$q_j = \sqrt[m]{\prod_{i=1}^{m} a_{ij}} / \sum_{j=1}^{n} \left( \sqrt[m]{\prod_{i=1}^{m} a_{ij}} \right) \tag{4}$$

Finally, the consistency is checked.

② The Entropy Weight Method derives the weight of each index based on the degree of disorder present in the index data [44,45]. The steps are as follows:

First, entropy is defined. It is assumed that there are a total of $m$ research objects (provinces, autonomous regions, and municipalities) and $n$ indicators; $x_{ij}'$ is the standardized value of the $j$th index of the $i$th research object. Then, the entropy of the $j$th index is:

$$e_j = -\frac{\sum\limits_{i=1}^{m} p_{ij} \ln p_{ij}}{\ln m} \tag{5}$$

where $p_{ij} = x_{ij}' / \sum\limits_{i=1}^{m} x_{ij}'$; when $p_{ij} = 0$, let $p_{ij} \ln p_{ij} = 0$.

The entropy weight is defined. The entropy weight of the $j$th index is:

$$h_j = d_j / \sum_{j=1}^{n} d_j \tag{6}$$

where $d_j = 1 - e_j$.

The additive integration method was employed to calculate the combined weight:

$$w_j = \alpha q_j + \beta h_j \tag{7}$$

where $w_j$ is the combined weight, and $q_j$ and $h_j$ are subjective and objective weight respectively. $\alpha$ and $\beta$ are the adjustment coefficients. In this article, subjective and objective weights are equally important, so take $\alpha = \beta = 0.5$.

(3) Comprehensive evaluation:

The following formula is used to evaluate forest ecological security:

$$P_i(S_i, R_i) = \sum_{j=1}^{n} w_j x'_{ij} \tag{8}$$

$$FES_i = (P_i \times S_i \times R_i) \tag{9}$$

In Formula (8), $x'_{ij}$ is the standardized index value, and $w_j$ is the index weight. $P$, $S$, and $R$ are the pressure, state, and response indices of forest ecological security, respectively. In Formula (9), FES is a comprehensive index of forest ecological security, which falls between 0 and 1. The larger the value of FES, the higher the level of forest ecological security. In order to facilitate the evaluation and analysis of forest ecological security, this study, referring to the existing relevant studies [46–48], divided forest ecological security into five levels: FES $\in$ [0, 0.2), severe level; FES $\in$ [0.2, 0.4), sensitive level; FES $\in$ [0.4, 0.6), critical safe level; FES $\in$ [0.6, 0.8), relatively safe level; and FES $\in$ [0.8, 1], safe level.

### 2.2.2. Coupling Coordination Degree Model

Coupling coordination analysis is beneficial for promoting the system from disorder to order and achieving coordinated development step-by step [49]. The formulas are as follows:

$$C = \left\{ \frac{P \times S \times R}{[(P + S + R)/3]^3} \right\}^{1/3} \tag{10}$$

where $P$, $S$, and $R$ are the pressure, state, and response indices of forest ecological security, respectively. $C$ is the coupling degree, which is usually divided into four stages: $0 < C \leq 0.3$ is the low coupling stage; $0.3 < C \leq 0.5$ is the antagonistic stage; $0.5 < C \leq 0.8$ is the running-in stage; and $0.8 < C \leq 1$ is high coupling stage.

The coupling degree can only reflect the interaction intensity between the pressure, state, and response, but not the coordination degree among them [50]. Pseudo-high coupling occurs when the pressure, state, and response indices are all low. Therefore, the coupling coordination degree model is constructed to reflect the coordination development level among pressure, state, and response:

$$D = \sqrt{C \times T} \tag{11}$$

$$T = \alpha P + \beta S + \gamma R \tag{12}$$

where $D$ is the coupling coordination degree and $T$ is the comprehensive harmonic coefficient of pressure, state, and response, reflecting the overall coordination effect of the three. $\alpha$, $\beta$, and $\gamma$ are undetermined coefficients. It is believed that they are equally important; thus, $\alpha = \beta = \gamma = 1/3$. In order to determine the coordinated development stage of forest ecological security, the evaluation criteria in Table 2 were formulated by referring to previous relevant studies [51] and combining their methods with the actual characteristics of this study.

**Table 2.** Classification standard of coupling coordination degree of pressure, state, and response.

| Coupling Coordination Degree (D) | Coupling Coordination Level | Coupling Coordination Phase |
|---|---|---|
| [0–0.1) | Extreme incoordination | |
| [0.1–0.2) | Serious incoordination | |
| [0.2–0.3) | Moderate incoordination | Unacceptable |
| [0.3–0.4) | Mild incoordination | |
| [0.4–0.5) | Near incoordination | Transition |
| [0.5–0.6) | Barely coordinated | |
| [0.6–0.7) | Primary coordination | |
| [0.7–0.8) | Intermediate coordination | Acceptable |
| [0.8–0.9) | Good coordination | |
| [0.9–1.0] | High-level coordination | |

### 2.2.3. Exploratory Spatial Data Analysis (ESDA)

ESDA is used to analyze spatial dependence, spatial correlation, or spatial autocorrelation phenomena [52]. In this study, the global Moran index was used to estimate the spatial correlation of the pressure–state–response coupling coordination degree of forest ecological security:

$$I = \frac{\sum\limits_{i=1}^{n} \sum\limits_{j=1}^{n} w_{ij}(x_i - \overline{x})(x_j - \overline{x})}{\sigma^2 \sum\limits_{i=1}^{n} \sum\limits_{j=1}^{n} w_{ij}} \tag{13}$$

### 2.2.4. Spatial Econometric Model

Compared with the traditional econometric model, the spatial econometric model takes spatial factors into account and analyzes the influence of the spatial effects, which can be used to explore the spatial interactions between different geospatial units [53]. Because the types of forest ecological security in different regions are not independent of each other in space but have certain spatial correlations and spatial dependences, this study used a spatial econometric model to analyze the influencing factors of the pressure, state, response, and coordinated development of forest ecological security.

(1)   Spatial Panel Lag Model (SPLM)

The Spatial Panel Lag Model describes substantial spatial correlation, and the spatial effect mainly exists in the model in the form of explained variable lag. The formula is:

$$Y_{it} = \rho \sum_{j=1}^{n} W_{ij} Y_{jt} + \beta X_{it} + \mu_i + \nu_t + \varepsilon_{it} \tag{14}$$

(2)   Spatial Panel Error Model (SPEM)

The Spatial Panel Error Model describes spatial disturbance correlation, and the spatial effect mainly exists in the model in the form of error lag. The formula is:

$$Y_{it} = \beta X_{it} + \mu_i + \nu_t + \mu_{it} \tag{15}$$

$$\mu_{it} = \lambda \sum_{j=1}^{n} W_{ij} \mu_{it} + \varepsilon_{it} \tag{16}$$

(3)   Spatial Panel Durbin Model (SPDM)

The Spatial Panel Durbin Model integrates the spatial effects of both the dependent variables and the independent variables into the model. The formula is:

$$Y_{it} = \rho \sum_{j=1}^{n} W_{ij} Y_{jt} + \beta X_{it} + \theta \sum_{j=1}^{n} W_{ij} X_{it} + \mu_i + \nu_t + \varepsilon_{it} \tag{17}$$

In Formulas (14)–(17), $\rho$ and $\theta$ are the spatial regression coefficients of the explained variable and the explanatory variable, respectively. $\beta$ is the regression coefficient of the explanatory variable; $\lambda$ is the spatial regression coefficient of the error term; and $\mu_i$ and $\nu_t$ are the spatial fixed effect and time fixed effect, respectively. $\varepsilon_{it}$ is the random error term subject to independent identical distribution, and $W_{ij}$ is the spatial weight matrix. When $\theta = 0$ and $\rho \neq 0$, SPDM degenerates to SPLM; when $\theta + \rho\beta = 0$, SPDM degenerates into SPEM.

### 2.3. Data Sources and Processing

The data which we utilized were obtained from various statistical yearbooks. Among them, the *China Statistical Yearbook* is a kind of information annual published by the National Bureau of Statistics. It comprehensively reflects the development of China's economy and society and is the most comprehensive and authoritative statistical yearbook in China. The *China Forestry Statistical Yearbook* is compiled based on forestry statistical annual reports and

other relevant data. It has been published annually since 1987, including six parts: forest and wetland resources, ecological construction, industrial development, employees and labor remuneration, forestry investment, and forestry education. The *China Environmental Statistics Yearbook* is compiled by the State Bureau of Statistics and the Ministry of Environmental Protection, and is published annually. It covers 12 sections related to resources and environment, such as natural ecology, land use, forestry, etc. *The National Forest Resources Inventory* is organized by the competent forestry department under the State Council and is carried out in provinces, autonomous regions, and municipalities directly under the Central Government, as well as in large forest areas. From the founding of the People's Republic of China to 2020, China has conducted nine national forest resource inventories. In summary, all data sources used in this study were compiled by national government departments and are open to the public. Therefore, we believe that they are highly authoritative and reliable.

It should be noted that: (1) The total forestry output value of Heilongjiang includes that of Daxing 'anling; (2) to avoid the impact of price changes, all price data used in the article are comparable price indices (with 2006 as the reference year); and (3) the spatial weight matrix for spatial effects was derived from the reciprocal matrix of spatial Euclidean distances.

## 3. Results and Discussion

### 3.1. Spatial–Temporal Features of FES

Figure 1 shows the spatial distribution of each index in 2020 and its trend from 2006 to 2020. The results of the forest ecological security pressure index (for visual representation of the pressure level, the values 1-P are shown here because the pressure indicator data were reversed during the standardization process). Figure 1a showed that in 2020, the forest ecosystems in central and western regions were generally under lower pressure than those in the eastern region. Specifically, in 2020, the pressure index of Beijing and Hainan (in the eastern region), Inner Mongolia (in the central region) and Sichuan, Yunnan, Tibet, Gansu, and Qinghai (in the western region) ranged from 0 to 0.2, indicating that the forest ecosystem in the above regions faced less external pressure and that there was a weak impact from anthropogenic activities on the forest ecosystems. The abovementioned areas are mostly central and western provinces with low population densities and relatively low levels of economic development. Their economic and social development consumes fewer forest resources and produces lower amounts of pollutants from industrial production. Therefore, their forest ecosystems face less economic, social, resource, and environmental pressure. The pressure indices of 16 provinces (Shanxi, Jilin, Zhejiang, etc.) were between 0.2 and 0.4, which is in the middle. The pressure indices of seven regions, including Shandong, Tianjin, Shanghai, Hebei, Jiangsu, Anhui, and Henan, were between 0.4 and 0.6, which is relatively high. Overall, though, the pressure indices of 31 provinces in China were below 0.6. In terms of the changing trend of the pressure index, except for Beijing, Shanghai, Zhejiang, and Xizang, the pressure indices of all of the other provinces showed small declines, indicating that with the improvement of the economic and social development levels, the pressure of the forest ecosystem in most provinces is gradually increasing.

Figure 1b showed that in 2020, the forest ecological security state index of Tianjin was the lowest, and the state of the forest ecosystem was poor. On the one hand, the forest coverage rate (12.07%) and the proportion of natural forest (4.84%) in Tianjin are both very low; thus, the quantity and quality of its forest resources obviously lag behind those of other areas. On the other hand, the incidence rate of forest diseases and insect pests in Tianjin (37.43%) is the highest in China—almost 10 times the national average (3.79%)—which affects the health of the forest ecosystem in this region. The state indices of 9 provinces, including Hebei, Shanxi, and Shanghai, were between 0.2 and 0.4, and those of 12 provinces, including Beijing, Liaoning, and Anhui, were between 0.4 and 0.6. The states of the forest ecosystems in these regions were in the middle. Nine provinces, including Heilongjiang, Sichuan, and Yunnan, had higher state indices (between 0.6 and 0.8), indicating forest ecosystems in better states. Most of the abovementioned nine provinces are located in the northeast forest area, the southwest forest area, and east China. Most of their forest

coverage exceeds 40%, the proportion of natural forest exceeds 60%, the stock volume of forest per hectare exceeds 70 cubic meters, and the incidence of forest diseases and insect pests is significantly lower than in other regions. Therefore, the forest ecosystems in the abovementioned areas are in good condition. In terms of the changing trends of the state indices, all provinces except Inner Mongolia, Jiangxi, and Guangdong showed slight declines in their state indices, indicating that the states of forest ecosystems in most provinces in China are gradually improving.

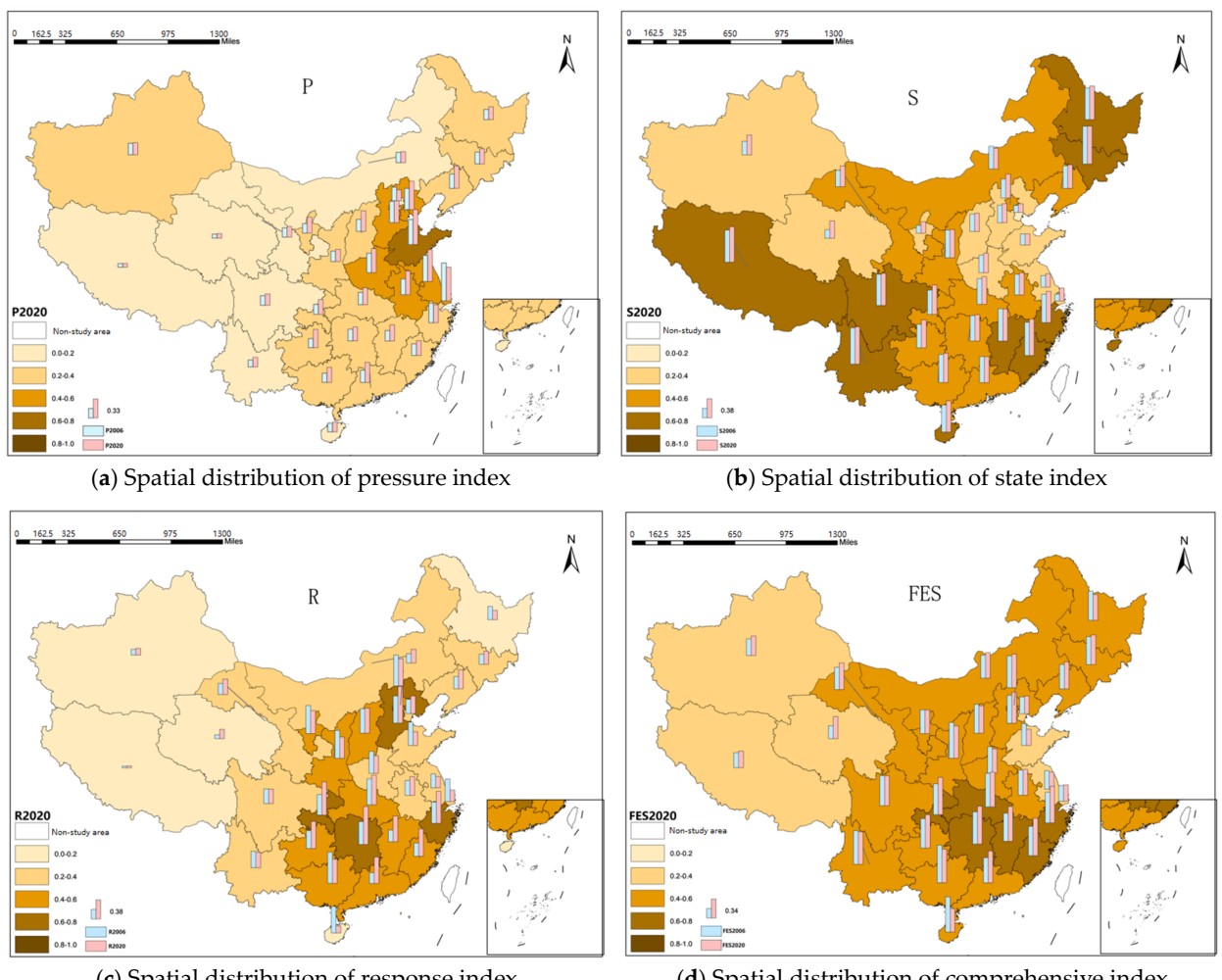

(**a**) Spatial distribution of pressure index

(**b**) Spatial distribution of state index

(**c**) Spatial distribution of response index

(**d**) Spatial distribution of comprehensive index

**Figure 1.** Spatial distribution of forest ecological security pressure, state, response, and comprehensive index.

The outcomes of the forest ecological security response index, as depicted in Figure 1c, revealed that in 2020, the response indices of Heilongjiang, Hainan, Tibet, Qinghai, and Xinjiang ranged from 0 to 0.2, and the protection of forest ecosystems by humans was relatively insignificant. The response indices of 12 provinces, including Tianjin, Inner Mongolia, and Liaoning, and 10 provinces, including Beijing, Shanxi, and Fujian, ranged from 0.2 to 0.4 and from 0.4 to 0.6, respectively, and the response level in terms of forest ecological security was in the middle. The response indices of Hebei, Zhejiang, Hunan, and Chongqing ranged from 0.6 to 0.8, indicating that these regions had relatively strong responses to needs of the forest ecosystems. In terms of the changing trend of the response indices, the response indices of 12 provinces, including Beijing, Heilongjiang, and Shanghai declined. Among them, the response index of Hainan declined the most because the proportion of new afforestation, mountains being closed for forest cultivation, and the utilization rate of industrial solid waste in Hainan showed sharp declines, which led to a

sharp decline in the response index of its forest ecological security. The decline in the rest of the region was relatively small.

The outcomes of the forest ecological security comprehensive index, as depicted in Figure 1d, revealed that in 2020, the forest ecological security of most provinces were deemed critical and relatively safe. Specifically, the comprehensive indices in seven provinces (Tianjin, Shanghai, Jiangsu, and Shandong in the east; Tibet, Qinghai, and Xinjiang in the west) were between 0.2 and 0.4 at the sensitive level. Because most areas in the eastern region mentioned above have relatively high levels of economic development, their forest ecosystems face greater economic and social pressures. However, the forest resources in western provinces are often inferior to other provinces due to the influence of natural conditions. Limited by regional economic level, their response measures to forest ecosystems are obviously insufficient. The combination of many factors eventually led to a lower level of forest ecological security in the abovementioned areas. The comprehensive indices in Zhejiang, Fujian, and Chongqing in the eastern region and Jiangxi, Hubei, and Hunan in the central region ranged from 0.6 to 0.8, which is the relatively safe level, while those of the remaining provinces reached the critical safety level. In terms of the changing trend, the comprehensive indices of forest ecological security in Hainan, Shandong, Guangxi, and other 10 provinces decreased. Among them, Hainan showed the largest decline, which was caused by the sharp decline in its response index according to the above analysis. The declines in other provinces were all less than 0.1. The comprehensive indices in 21 provinces, including Beijing, Zhejiang, and Hunan, showed upward trends. Among them, Zhejiang, Fujian, Jiangxi, Hunan, Guangdong, Chongqing, and Qinghai had increases of 0.1–0.2, while the other provinces had increases of less than 0.1. In summary, the forest ecological security in the majority of provinces exhibited a consistent improvement from 2006 to 2020, and even though there were downward trends in a few provinces, the declines were small.

### 3.2. Spatial–Temporal Features of FESD

The coupling degree of pressure–state–response (Figure 2) showed that in the study period, the coupling degrees of 28 provinces were always higher than 0.8, which is in the high coupling stage, accounting for 90.32% of the total number of regions. The coupling degrees of Qinghai and Xinjiang were always between 0.5 and 0.8 in the running-in stage. Tibet was in the antagonistic stage (2006), low coupling stage (2013), and running-in stage (2020), respectively. The reason for this is that during the study period, Tibet's forest ecological security pressure index and state index were basically stable, while the response index showed a declining trend first and then a rise, which led to the same changing trend in the coupling degrees of the three. In terms of the changing trend, although none of the coupling stage of the regions (except Tibet) changed from 2006 to 2020, the coupling degrees of all other regions except Hebei, Heilongjiang, Hainan, Yunnan, and Shaanxi showed upward trends. Moreover, the number of provinces with coupling degrees above 0.9 increased from 20 in 2006 to 24 in 2013 and 25 in 2020.

The coupling coordination degree of pressure–state–response (Figure 3) showed that, in general, during the study period, the coupling coordination degrees of all provinces were higher than 0.4, except for Tibet in 2013, which was under 0.4 (in the stage of moderate incoordination). Among them, in 2006, one province (Qinghai) was at the stage of near incoordination, and five provinces (Xinjiang, Tibet, Jiangsu, Shanghai, and Tianjin) were at barely coordinated stage. The pressure–state–responses of the above six provinces were in the transitional interval of coordinated development. There were 8 (Inner Mongolia, Liaoning, Anhui, etc.) and 16 (Shaanxi, Yunnan, Sichuan, etc.) provinces in the primary and intermediate stages of coordination, respectively. Guangxi had the highest degree of coupling coordination and was at the stage of good coordination. The pressure–state–responses of the above 25 provinces were in the acceptable range of coordinated development, accounting for 80.65% of the total number of regions. In 2013, there were still 6 provinces in the transition range and 24 provinces in the acceptable range. Among

them, the number of provinces in the good coordination stage increased to three (Fujian, Chongqing and Yunnan). In 2020, provinces in the near incoordination stage no longer existed. Six provinces were in the barely coordination stage (transitional interval), while 9 provinces (Inner Mongolia, Liaoning, Heilongjiang, etc.) and 12 provinces (Beijing, Hebei, Shanxi, etc.) were at the primary and intermediate coordination stages, respectively. The number of provinces at the good coordination stage was further increased to four (Zhejiang, Fujian, Hunan, Chongqing). In conclusion, during the study period, the coupling and coordinated development levels in most provinces were in acceptable ranges and showed trends of continuous improvement.

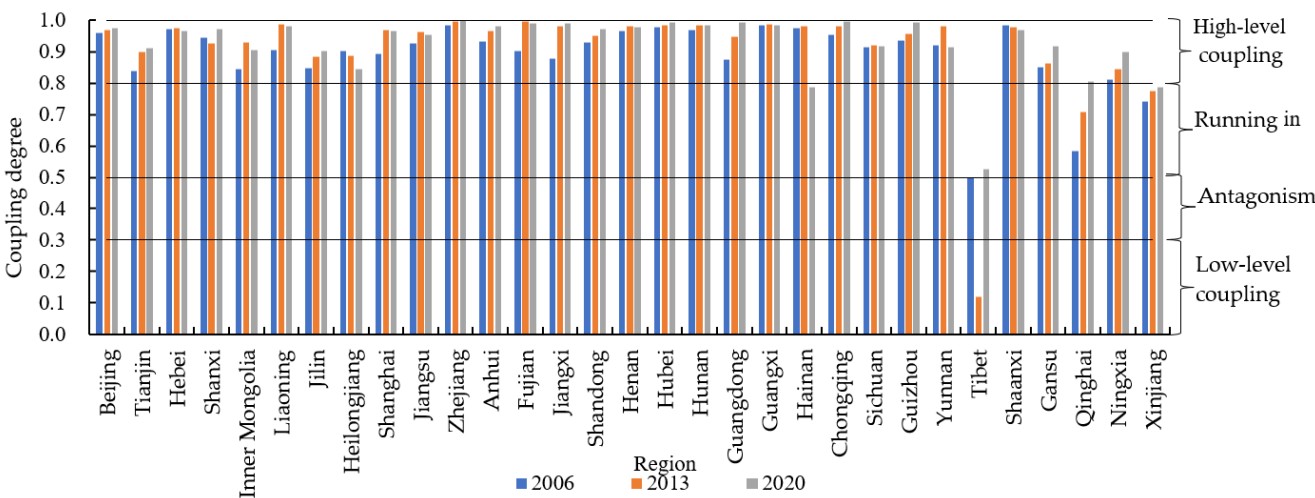

**Figure 2.** Coupling degree of pressure–state–response from 2006 to 2020.

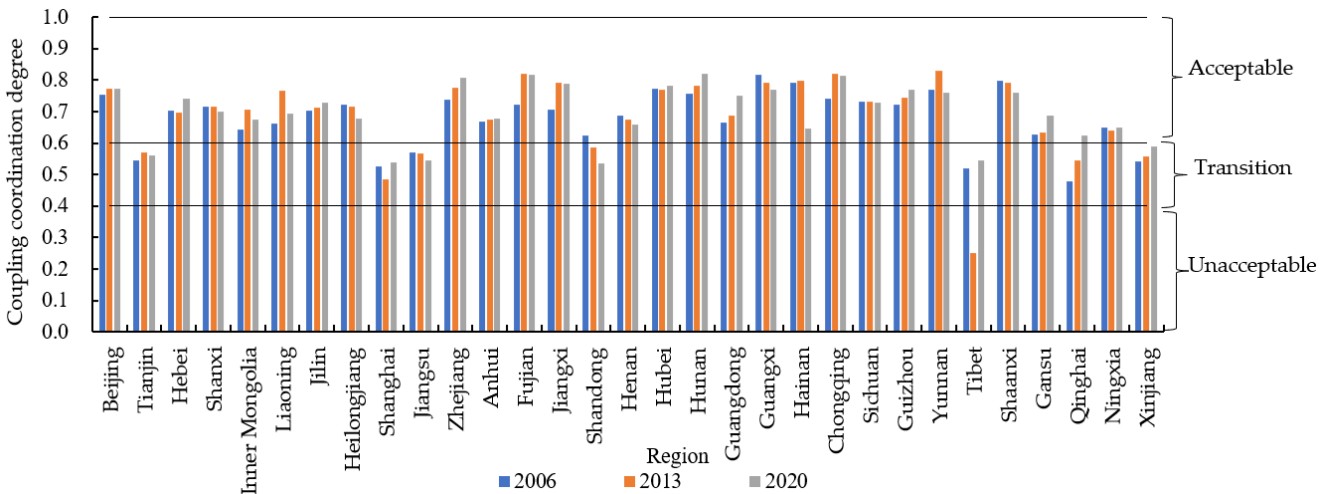

**Figure 3.** Coupling coordination degree of pressure–state–response from 2006 to 2020.

### 3.3. Spatial Influencing Factors of FESD

The coordinated development of the pressure–state–response of forest ecological security is affected by many factors. This study comprehensively considers the interaction relationship of the pressure–state–response and the availability of data, selecting influencing factors from nature, economic society, science, technology, and education.

Natural factors: Both the theory of industrial location and the theory of regional comparative advantage show that natural factors are the premise of production. They have an important impact on economic development, industrial layout, and population layout [54,55], and they determine the demand for forest resources and pollution emissions of a region, i.e., the pressure on the forest ecosystem. Simultaneously, temperature,

precipitation, light, and other natural conditions are necessary factors for the growth of trees, which directly impact the quantity and quality of forest resources, that is, the state of forest ecosystem. The annual mean temperature, annual precipitation, and annual sunshine duration were selected to reflect natural conditions, but they were multicollinear, so only the annual mean temperature ($X_1$) was analyzed.

Socio-economic factors: On the one hand, population size, economic development, and urbanization level directly determine the destructive intervention (pressure) faced by the forest ecosystem, thereby changing the state of the forest ecosystem. On the other hand, the acceleration of economic development will promote forestry investment, thereby strengthening the protection of forest ecosystems (response). At the same time, economic development will also promote regional environmental governance, thus reducing the pressure on the forest ecosystem. Five indicators, including forest population density ($X_2$), urbanization rate ($X_3$), per capita GDP ($X_4$), completed forestry investment ($X_5$), and investment in environmental governance ($X_6$) were selected.

Technology and education factors: Technology and education influence the pressure and state of forest ecological security by changing the efficiency of forestry production and the management levels of forests. Forest pest control ($X_7$) and staff education level ($X_8$) were selected.

A spatial econometric model was used to explore the influencing factors and spatial spillover effects of FESD. Before the spatial econometric analysis, it was necessary to check whether the research object had significant spatial correlation. With the support of GeoDa, the Moran index of FESD was calculated to explore its spatial correlation (Table 3). The results showed that the Moran indices were all greater than 0 from 2006 to 2020, and all passed the test at the significance level of 5%, indicating that there was a significant positive spatial autocorrelation between the pressure–state–response coupling coordination degree of forest ecological security in the provinces and the exogenous interaction between explanatory variables. In view of this, the spatial econometric model was selected to estimate the spatial correlation of provinces using Stata.

**Table 3.** Moran indices of coupling coordination degrees of pressure–state–responses.

| Year | Moran Indices | *p*-Values | Z-Values |
|------|---------------|-----------|----------|
| 2006 | 0.3008 | 0.005 | 2.8188 |
| 2007 | 0.2932 | 0.006 | 2.8095 |
| 2008 | 0.2721 | 0.008 | 2.6776 |
| 2009 | 0.2335 | 0.013 | 2.406 |
| 2010 | 0.2540 | 0.009 | 2.5771 |
| 2011 | 0.2164 | 0.015 | 2.3328 |
| 2012 | 0.2846 | 0.008 | 2.7029 |
| 2013 | 0.2414 | 0.009 | 2.5758 |
| 2014 | 0.2696 | 0.008 | 2.632 |
| 2015 | 0.2575 | 0.008 | 2.6221 |
| 2016 | 0.2713 | 0.008 | 2.6384 |
| 2017 | 0.3501 | 0.002 | 3.1507 |
| 2018 | 0.3479 | 0.003 | 3.1177 |
| 2019 | 0.3586 | 0.001 | 3.2889 |
| 2020 | 0.3211 | 0.004 | 2.9456 |

A series of tests should be carried out for the selection of appropriate models. Table 4 shows that in the LM test, and both LM-lag and LM-error tests passed the test at the significance level of 1%. In the Robust LM test, LM-error did not pass the significance test, and LM-lag passed the test at a significance level of 1%. The aforementioned findings suggest the potential coexistence of a spatial lag effect and a spatial error effect, making the SPDM the preferred choice. At this time, it is also necessary to test the SPDM to check whether it can be degraded into the SPLM or the SPEM. Because both the Wald test and the LR test reject the null hypothesis that the SPDM can be reduced to the SPLM or the SPEM

at a significance level of 1%, the SPDM should be selected. Furthermore, the Hausman test rejects the null hypothesis at the 1% significance level, indicating the superiority of the fixed-effect model over the random-effect model. As a result, the fixed-effect SPDM was ultimately adopted in this study.

**Table 4.** Test results of the spatial econometric model.

| Method | | Results |
|---|---|---|
| Lagrange multiplier (LM) test | LM—lag | 7.301 *** |
| | LM—Robust lag | 13.408 *** |
| | LM—Error | 8.883 *** |
| | LM—Robust Error | 1.587 |
| Wald test | Wald—lag | 86.0692 *** |
| | Wald—error | 96.7233 *** |
| Likelihood ratio (LR) test | LR—lag | 83.1092 *** |
| | LR—error | 80.9027 *** |

Note: *** Significant at the 0.01 level.

Comparing the values of Log likelihood and R-squared, it was found that the SPDM had the best fitting effect, and the SPLM was slightly better than the SPEM (Table 5). Lesage et al. proposed that using the point estimation method set by the SPDM to verify whether there is spatial spillover effect may have some bias, that is, the coefficient estimate of the independent variable does not represent the true partial regression coefficient. As a result, the SPDM of space by means of partial differential effect decomposition has a direct effect according to the region. The explained variable is the region's FESD effect, an indirect effect of the region's various explanatory variables on the FESD neighborhood. The total effect is the sum of the direct and indirect effects of the explanatory variables of FESD's overall impact.

**Table 5.** Results of spatial panel models.

| Variables | Name | Spatial Panel Lag Model (SPLM) | Spatial Panel Error Model (SPEM) | Spatial Panel Durbin Model (SPDM)-Was Adopted | | | | |
|---|---|---|---|---|---|---|---|---|
| | | | | X | $W \times X$ | Direct Effect | Indirect Effect | Total Effect |
| $X_1$ | Annual mean temperature | 0.0094 *** [15.4552] | 0.0100 *** [15.5667] | 0.0076 *** [7.2015] | 0.0085 * [1.675] | 0.0077 *** (7.8906) | 0.0126 * (1.9961) | 0.0203 *** (3.5998) |
| $X_2$ | Forest population density | −0.0011 *** [−17.0613] | −0.0011 *** [−17.2140] | −0.0010 *** [−14.9102] | 0.0013 ** [2.1592] | −0.0010 *** (−13.8688) | 0.0014 * (1.8040) | 0.0004 (0.4632) |
| $X_3$ | Urbanization rate | 0.0046 *** [11.4530] | 0.0048 *** [10.7187] | 0.0067 *** [12.9798] | 0.0077 *** [3.3967] | 0.0068 *** (13.1898) | 0.0114 *** (3.4162) | 0.0183 *** (5.3836) |
| $X_4$ | Per capita GDP | −0.0601 *** [−6.5623] | −0.0740 *** [−6.2271] | −0.1365 *** [−8.0586] | −0.0017 [−0.0340] | −0.1367 *** (−7.8807) | −0.0398 (−0.6151) | −0.1764 *** (−2.7991) |
| $X_5$ | Completed forestry investment | 0.0357 *** [11.3808] | 0.0353 *** [11.3681] | 0.0387 *** [12.6438] | 0.0448 *** [2.636] | 0.0395 *** (12.3753) | 0.0662 *** (3.1296) | 0.1056 *** (4.7467) |
| $X_6$ | Investment in environmental governance | −0.0002 *** [−6.0324] | −0.0002 *** [−6.4145] | −0.0002 *** [−6.7655] | 0.0001 [0.5401] | −0.0002 *** (−6.7436) | 0.00004 (0.2287) | −0.0002 (−0.9467) |
| $X_7$ | Forest pest control | 0.0005 *** [3.6455] | 0.0006 *** [3.9845] | 0.0003 * [1.8534] | −0.0050 *** [−5.0830] | 0.0002 (1.4041) | −0.0062 *** (−4.6122) | −0.0060 *** (−4.2906) |
| $X_8$ | Staff education level | 0.0006 *** [2.7207] | 0.0006 *** [2.6386] | 0.0003 [1.4836] | −0.0007 [−0.5150] | 0.0003 (1.4699) | −0.0007 (−0.4171) | −0.0004 (−0.2242) |
| $\rho$ & $\lambda$ | | 0.3990 *** [5.0928] | 0.4920 *** [5.7605] | 0.1930 [1.6312] | | | | |
| Constant | | 0.1718 * (1.6830) | 0.5790 *** (4.9914) | 0.2686 (0.7464) | | | | |
| R-squared | | 0.6384 | 0.6164 | 0.6952 | | | | |
| sigma$^2$ | | 0.0038 | 0.0037 | 0.0032 | | | | |
| Log likelihood | | 634.8076 | 635.8648 | 676.3823 | | | | |

Note: All the models are mixed-effect spatial panel models; [] is asymptotic t-statistics, () is t-statistics. *** Significant at the 0.01 level, ** significant at the 0.05 level, * significant at the 0.10 level.

Annual mean temperature, urbanization rate, completed forestry investment, and forest pest control have positive influences on FESD. It is indicated that the above factors exert a favorable influence on alleviating the pressure on forest ecosystems, improving the state of forest ecosystems, and enhancing the response to forest ecological security needs. For example, temperature is a necessary condition for plant growth, and appropriate temperature is conducive to promoting the growth of trees, which can increase the amount of forest resources. Regional urbanization and technological development often complement each other. Science and technology exert a favorable influence in forest ecological management and forest resource protection. Forestry investment can not only improve the protection of forest ecosystems but also strengthen the management of forest ecosystems. Forest pest control promotes FESD mainly by improving the health of forest ecosystems.

Forest population density, per capita GDP, and investment in environmental governance have significant negative influences on FESD. The above factors have negative effects on FESD, mainly by increasing the pressure on the forest ecosystem. For example, the greater the forest population density, the greater the consumption of forest resources and the occupation of forest land, which will increase the resource pressure on the forest ecosystem. A higher per capita GDP tends to be associated with more pollution emissions, and more investment in environmental treatment tends to mean more regional pollutants, which will increase the environmental pressure on forest ecosystems.

The ranking of direct effects is as follows: Per capita GDP($-$) > Completed forestry investment > Annual mean temperature > Urbanization rate > Forest population density($-$) > Investment in environmental governance($-$), ("$-$" represents a negative effect). The direct effects of forest pest control and staff education on FESD were not significant. The order of spatial spillovers is as follows: Completed forestry investment > Annual mean temperature > Urbanization rate > Forest pest control($-$) > Forest population density. Most of the above factors show spatial correlation or spatial fluidity. For example, the spatial externality theory holds that regional investment has a significant spillover effect. Although the implementation scope of local investment is limited to a certain region, its impact is not limited to this region. It can be inferred that forestry investment in a certain province will not only change the local FESD but also have an impact on other provinces. Temperature is a necessary condition for the growth of trees, and the temperature of an area is often similar to that of its surrounding areas due to geographical proximity, so temperature is the main factor affecting the spatial correlation of FESD. The occurrence of forest pests and diseases easily triggers chain reactions, resulting in spatial contiguous effects. The flow of people, capital, information, and technology between different regions is the main reason for the spatial spillover effects of urbanization and forest population density on FESD.

## 4. Conclusions, Enlightenment, and Prospects

The following conclusions were obtained through our analysis of the internal coupling coordination of forest ecological security and its spatial influencing factors.

(1) The adverse effects of human production and living activities on forest ecosystems in central and western China are weaker than those in eastern China. Except for Beijing, Shanghai, Zhejiang, and Tibet, the external disturbances faced by forest ecosystems in other provinces have shown a gradual increasing trend.

(2) The quantity, quality, and health of forest resources in northeast and southwest forests are better than those in other regions. Except Inner Mongolia, Jiangxi, and Guangdong, the states of forest resources in other provinces have shown a trend of continuous improvement.

(3) The response measures taken by Heilongjiang, Hainan, Xizang, Qinghai, and Xinjiang to protect forest ecosystems and maintain forest ecological security were relatively insufficient compared with Hebei, Zhejiang, Hunan, and Chongqing. Protection of the forest ecosystem and control of environmental pollution in 12 provinces, including Heilongjiang, Beijing, and Shanghai, has been gradually decreasing, while in other provinces these factors are increasing.

(4) The forest ecological security levels of most provinces in China are critical or relatively safe, and the forest ecological security of most provinces is constantly improving, except for provinces such as Hainan and Shandong. The external disturbance of forest ecosystem is increasing, but the level of forest ecological security is gradually improving, indicating that China's forest protection measures and environmental governance policies have significantly contributed to the enhancement of forest ecological security.

(5) Except for Qinghai, Xinjiang, and Tibet, forest ecological security is always in a highly coupled stage and has a trend of continuous improvement. The internal coordination degrees of forest ecological security in 25 provinces, represented by Guangxi, are in the acceptable range and show trends of gradual improvement.

(6) FESD is affected by the comprehensive factors of nature, economy, society, science and technology, education, etc. Among these, annual mean temperature, urbanization rate, completed forestry investment, and forest pest control have positive influences on FESD. Forest population density, per capita GDP, and investment in environmental governance have significant negative influences. In addition, completed forestry investment, annual mean temperature, urbanization rate, and forest population density have significant positive spillover effects on FESD, and forest pest control has a significant negative spillover effect on FESD.

The above research findings lead to the following implications:

(1) The results show that the forest ecosystem in most provinces cannot bear the human consumption of forest resources and the destruction of forest ecology, which leads the forest ecosystem into a state of unsustainable development. Hence, during forest ecological management, it is essential to not only focus on the condition of the forest ecosystem, but also to reduce the pressure on it by reducing woodland occupation and controlling pollution discharge. In addition, efforts should be made to enhance the response level of forest ecological security, increase the investment in environmental pollution control and forestry, and pay attention to the management and suppression of forest fires and insect pests.

(2) When the forest ecological security level of a region reaches a certain stage, it is impossible to achieve sustainable improvement simply by relying on its own efforts. At this time, the spatial spillover effect among regions can be harnessed to elevate the overall level of forest ecological security. The first law of geography posits that all things are interconnected, and proximity in distance corresponds to a closer relationship. Therefore, in the process of creating a series of policies, such as economic and social development, urbanization construction, and ecological environmental governance, the mutual influence between regions should be fully considered in order to reduce the regional differences in forest ecological security and promote the progressive enhancement of China's forest ecological security from an individual to a local level, and then to the whole of the country.

Drawing on the outcomes of this investigation, the subsequent research directions and questions can be further discussed and expanded on in the future:

(1) Forest ecological security is comprehensively affected by forest resources, the forest ecological environment, economic and social development, forest ecological policies, and other factors, so there will inevitably be shortcomings in the construction of an index system for forest ecological security evaluation. Follow-up studies can improve the evaluation index system of forest ecological security by strengthening the theoretical understanding of forest ecological security.

(2) When studying the coupling and coordination relationships of each subsystem in forest ecological security, social network analysis can be used to further explore the spatial correlation characteristics and driving mechanisms of the coupling and coordination degrees of forest ecological security from the perspective of a spatial correlation network.

**Author Contributions:** Conceptualization and funding acquisition, X.C. and J.L.; methodology, data collection and analysis, X.C., Z.S. and B.Z.; writing—review and editing, X.C., Z.S. and T.Y. All authors have read and agreed to the published version of the manuscript.

**Funding:** This research was jointly funded by the Fundamental Research Funds for the Central Universities (grant number 2572022DE02), and National Social Science Foundation (grant number 21BGL166).

**Data Availability Statement:** The data used in the study comes from paper statistical yearbooks, please refer to Section 2.3 for details.

**Conflicts of Interest:** The authors declare no conflict of interest.

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
