# Peer review of "Identifying the Internal Coupling Coordination Relationship of Forest Ecological Security and Its Spatial Influencing Factors"

_forests, doi:10.3390/f14081670_

Round 1

Reviewer 1 Report

It is necessary to focus on the tense of the manuscript, and the existing research must be in the past tense.

Reviewer 2 Report

The abstract is a bit too technical for the public and I recommend you to remodel it in places so that the comprehensiveness generates more citations

Research methods and methodology are clearly detailed to be able to follow the applicability of the research

Following the presentation of the results from Table 5. Results of spatial panel models, insert a few sentences including explanations and sociological implications

Please expand the conclusions section

Please check if all bibliographic references are included and check spelling and grammar in places

Overall, the paper is well structured and the chosen research topic is interesting.

Please check if all bibliographic references are included and check spelling and grammar in places

Reviewer 3 Report

The manuscript deals with an interesting piece of work with an in-depth analysis of huge data, and I thank you for the hard work done by the Chinese authors. But the authors have ample scope to improve the manuscript, and here are my specific suggestions:

  1. I think the author should specify the objectives for better understanding by the reader, and there is no need to justify the objectives at the end of the Introduction section.
  2. At the end of the abstract, the authors need to make a general recommendation for worldwide readers based on their own findings.
  3. The main limitation of the manuscript is poor discussion, and authors can easily reduce the conclusion section by improving the discussion parts. Make the findings simpler and discuss your findings for the readers.
  4. I also suggest including a future direction for the readers in the conclusion section and summarizing the key findings precisely.

Good luck 

Round 2

Reviewer 1 Report

The reference format of this manuscript still needs some revision. The authors have addressed most of my concerns and I think this manuscript could be considered for publication, good luck!

Author Response

The reference format of this manuscript has been revised. Thank you for your valuable revision suggestions.

Reviewer 2 Report

The author responded clearly to all the reviewer comments. No further comments or explanations are required. 

No additional spelling aln grammar corrections  are necessary 

Author Response

Thank you for your valuable suggestions.